

# Recent dating of extinct Atlantic gray whale fossils, (*Eschrichtius robustus*), Georgia Bight and Florida, western Atlantic Ocean

Ervan G. Garrison[1], Gary S. Morgan[2], Krista McGrath[3], Camilla Speller[3,4] and Alexander Cherkinsky[5]

[1] Geology, University of Georgia, Athens, GA, USA
[2] New Mexico Museum of Natural History & Science, Albuquerque, NM, USA
[3] University of York, BioArch Centre, York, UK
[4] Department of Anthropology, University of British Columbia, Vancouver, BC, Canada
[5] Center for Applied Isotope Studies, University of Georgia, Athens, GA, USA

## ABSTRACT

The Atlantic gray whale (*Eschrichtius robustus*) presents an interesting case study of climate related dispersal and extinction. While (limited) fossil records confirm its presence in the Atlantic up until the 18th Century, its abundance and distribution within the Eastern and Western basins are still not well understood. The discovery of presumed gray whale fossil remains from the Georgia Bight and the Atlantic coast of Florida, from the mid-1980s to late-2000s, provides a new opportunity to recover additional data regarding their chronology within the Western basin. Here, we apply accelerator mass spectrometry radiocarbon techniques to six fossil whale finds, identifying dates within marine isotope stage 3 (59–24 ka) and the late Holocene, ~2,000 yr BP. We additionally confirm the taxonomic identification of two fossil bone samples as *E. robustus* using collagen peptide mass fingerprinting (ZooMS). The obtained dates, when combined with a larger corpus of previously published Atlantic gray whale fossil dates, support the hypothesis for the decline of the Atlantic gray whale in the late Pleistocene and the late Holocene. These new data augment the findings of the Eastern Atlantic Basin and better incorporate the Western Atlantic Basin into a pan-ocean understanding for the species.

# INTRODUCTION

Recent studies have suggested that changes in species distribution during the Late Pleistocene–Early Holocene transition were caused both by habitat tracking as well as by extirpation of populations outside of isolated habitat refugia (*Dalén et al., 2007*; *De Bruyn et al., 2011*; *Hofreiter et al., 2004*; *Hofreiter, 2008*; *Hofreiter & Stewart, 2009*; *Stewart, 2009*). The gray whale (*Eschrichtius robustus*) presents an interesting case study by which to study this type of climate-related dispersal and extinction. Based on fossil evidence, the gray whale was present in both the North Pacific and North Atlantic during

Corresponding authors
Ervan G. Garrison, egarriso@uga.edu
Camilla Speller,
speller@palaeome.org

the Late Pleistocene and early Holocene, and migration from the Pacific to the Atlantic is thought to have been strongly shaped by Pleistocene climate shifts affecting potential dispersal routes and benthic feeding habitats. Both fossil and historical accounts suggest that the Atlantic gray whale was extinct in the North Atlantic by the mid-1700's (*Mead & Mitchell, 1984*; *Lindquist, 2000*), with both climate and anthropogenic factors implicated in its demise. Nevertheless, due to a paucity of fossil data in both the Eastern and Western basins, the diachronic distribution and abundance of gray whales prior to their disappearance is not well understood.

The gray whale family (Eschrichtiidae) is one of four families within the Cetacea suborder Mysticeti. Taxonomically, the phylogenetic placement of Eschrichtiidae has been controversial, with several suggested topologies (*McLeod, Whitmore & Barnes, 1993*; *Arnason, Gretarsdottir & Widegren, 1992*; *Arnason & Gullberg, 1993*, *1994*, *1996*; *Sasaki et al., 2005*). Ironically, the extant gray whale *E. robustus* was originally described (as *Balaenoptera robusta*) based on a Holocene fossil skeleton from Gräsö, Sweden (*Lilljeborg, 1861*; English translation in *Lilljeborg, 1867*). *E. robustus* was discovered as a living animal in the Pacific shortly thereafter. The family Eschrichtiidae includes only one living genus and species, *E. robustus* but modern workers describe four putative genera of Eschrichtiidae: *Eschrichtius*, *Archaeschrichtius*, *Eschrichtioides*, and *Gricetoides* (*Bischonti, 2008*; *Deméré et al., 2008*; *Ichishima et al., 2006*; *Steeman, 2007*). As the North Atlantic gray whale population went extinct prior to formal analysis of its taxonomy (*Barnes & McLeod, 1984*; *Lindquist, 2000*), it was not clear if the Atlantic and Pacific populations represented distinct species. Nor is it clear what drove the extirpation of this species in the North Atlantic. *Lindquist (2000)* is by far the most authoritative review of this question particularly with regard to any anthropogenic causation—prehistoric and historic—such as whaling, either opportunistic procurement or systematic hunting. It is noteworthy that two separate archaeological studies using DNA and collagen peptide mass fingerprinting (PMF) (i.e. ZooMS) methods, respectively found no evidence of gray whale in historic whaling assemblages in the western North Atlantic (*McLeod et al., 2008*) or eastern North Atlantic (*Buckley et al., 2014*). Recent genetic comparisons of North Atlantic fossil material and extant North Pacific populations by *Alter et al. (2015)*, however, confirmed that they represent the same species, connected through intermittent inter-ocean exchange or dispersals from the Pacific during openings of the Bering Strait (*Alter et al., 2015*). Moreover, combined radiocarbon dating and genetic results led Alter et al. to conclude that "dispersal between the Pacific and Atlantic was climate-dependent, and occurred—at least twice—both during the Pleistocene prior to the last glacial period, and the early Holocene immediately following the opening of the Bering Strait."

In the Eastern Atlantic Basin, Pleistocene marine mammal taxa are known from the southern part of the North Sea and the Eurogeul—recovered during the recent channel deepening project of the Rotterdam Harbor. Based on morphology and radiocarbon dates, the fossils from the Eurogeul locality belong to a cold Late Pleistocene fauna (*Mol et al., 2006*), and confirm with certainty the Late Pleistocene as well as the Holocene occurrence of gray whales in the eastern Atlantic Ocean. In contrast, only a dozen fossils

have been recovered from the Western Basin of the Atlantic, with the majority of dated remains corresponding to Holocene (*Mead & Mitchell, 1984*; *Bryant, 1995*). Only one study to date has recorded a late Pleistocene presence in this region (*Noakes, Pyenson & McFall, 2013*), raising questions about the extent to which the Western Basin was frequented by gray whales prior to the Holocene, and the genetic relationship between Western and Eastern populations of Atlantic gray whale. To date, relatively few archaeological or paleontological gray whale specimens have been identified on the Eastern seaboard of North America, and ancient DNA (aDNA) surveys of historic whaling assemblages in this region have failed to detect any additional gray whale specimens (*Rastogi et al., 2004*; *McLeod et al., 2008*). This overall paucity of fossil and genetic data makes it difficult to draw meaningful conclusions about the abundance, distribution, and chronology of Atlantic gray whales in the Western Basin during both the Pleistocene and Holocene.

Here, we apply accelerator mass spectrometer (AMS) dating to five new fossil gray whale finds from Florida and Georgia, and confirm taxonomic identifications of two craniae using collagen PMF. Our results provide new evidence for the Late Pleistocene presence of gray whale in the Western basin, and combined with other fossil dates for the Atlantic, support the decline of Atlantic gray whales in the Pleistocene prior to post-glacial re-population.

## METHODS

In our study, we analyzed six fossil gray whale specimens from the western Atlantic Ocean. We newly applied AMS dating to three of the finds and collated these with ages of previously published specimens (*Garrison et al., 2012*). Collagen PMF (ZooMS) was performed to confirm taxonomic identifications based on standard morphological comparisons. Two skulls were found on Florida beaches (Fig. 1), one dentary, and three puted dentary fragments were recovered from an underwater site, offshore Georgia (*Noakes, Garrison & McFall, 2009*; *Garrison et al., 2012*; *Noakes, Pyenson & McFall, 2013*) (Figs. 1–3).

The gray whale fossils from the Gray's Reef and JY Reef localities, offshore Georgia, are cataloged in the paleontology collection of the Georgia Museum of Natural History (GMNH), University of Georgia, Athens. The fossils from the Hobe Sound and Jacksonville Beach localities in Florida are cataloged in the vertebrate paleontology collection of the Florida Museum of Natural History, University of Florida (specimen acronym UF), Gainesville.

### Fossil localities of the Georgia and Florida gray whales

The Georgia locality is J-Reef a low exposure of shell beds about 16 km north of the Gray's Reef (Fig. 1).

At J-Reef the shell beds are in a conformable relationship with finer grained sediment that dates to no earlier than the late Pleistocene (*Garrison et al., 2012*).

The dentary and three putative dentary fragments shown in Fig. 2 were recovered from the fine sediment or adjacent to the same outcrop on the sea floor. During subsequent dives, in fall 2006, the large fossil dentary was discovered partially embedded in the

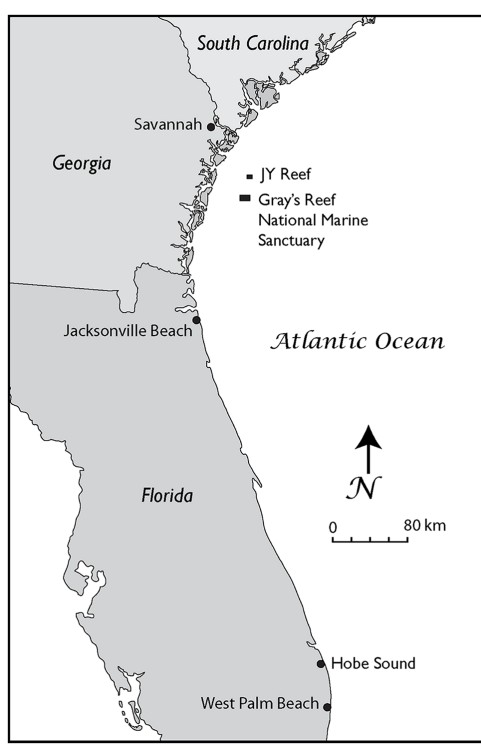

**Figure 1 The Georgia Bight and Florida Peninsula, showing the location of the gray whale specimens discussed in the text.** These records include specimens from the JY Reef and Gray's Reef National Marine Sanctuary in the Georgia Bight and Jacksonville Beach and Hobe Sound on the Atlantic Coast of Florida.

coquina (Fig. 2B). This coquina sand, typically deposited on a beach or in a nearshore marine environment, was probably the original sediment, like that of the Hobe Sound skull, in which it was preserved before being exposed.

The following details on the discovery and exact location of the gray whale skull from Hobe Sound (Fig. 3, Specimen UF 69000) were provided by Burkett S. Neely, Jr. (in litt., 6 June 1983) of the U. S. Fish and Wildlife Service (USFWS). The skull was discovered by a Mr. and Mrs. Kornit on 19 January 1983 at the edge of the surf on the northern end of Jupiter Island about 10 km south of St. Lucie Inlet, on the Hobe Sound National Wildlife Refuge (NWR), Martin County, southeastern Atlantic Coast of Florida (Fig. 1). Exact coordinates for the Hobe Sound skull are 27°07′40″ North latitude and 80°08′45″ West longitude. The skull was collected later that same day by USFWS personnel. The discovery of the Hobe Sound skull apparently coincided with extensive beach erosion that occurred during a strong winter storm. Through the generosity of Burkett S. Neely, Jr. and the USFWS, the Hobe Sound gray whale skull was placed on permanent loan in the vertebrate paleontology collection of the Florida Museum of Natural History (FLMNH), University of Florida (UF), Gainesville (catalogue number UF 69000). *Mead & Mitchell (1984*, 48*)* stated that "The specimen consists of most of the cranium of what looks like an adult . . ." However, upon closer examination, this skull consists only of a braincase, including both periotics (Fig. 3), and is from a very young individual or calf probably less than a year old. Robert K. Bonde

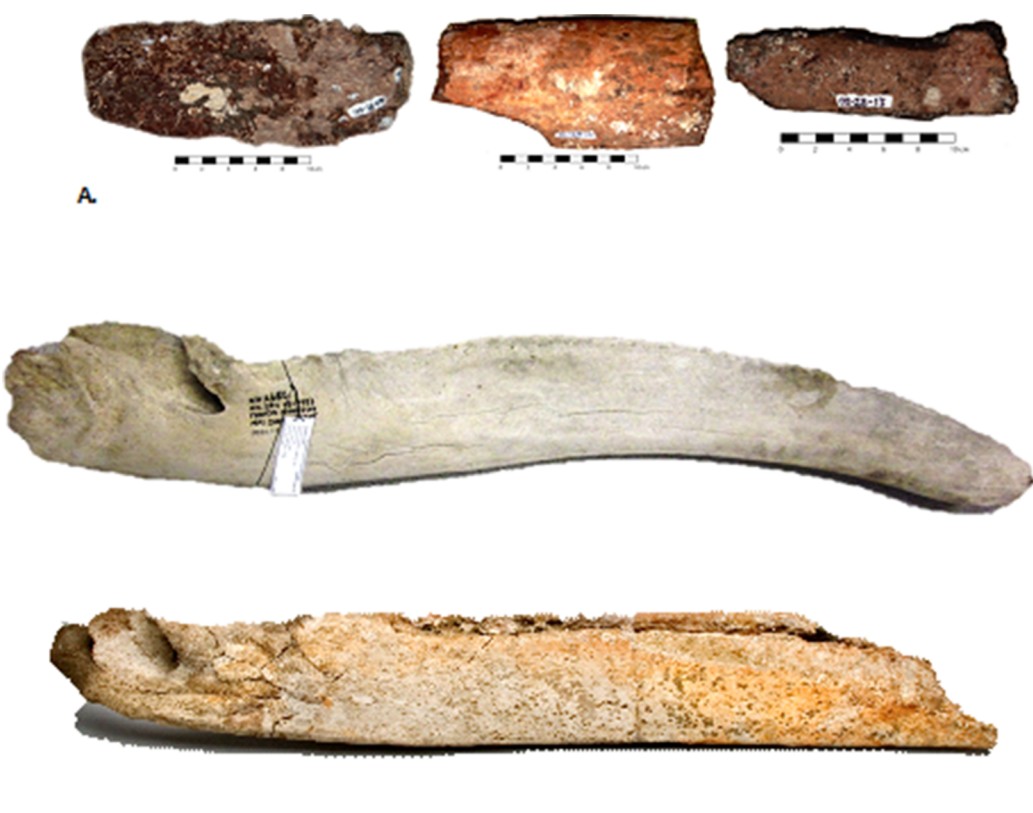

**Figure 2 The late Pleistocene and Holocene specimens of the gray whale (*Eschrichtius robustus*), Research Ledge site, JY Reef, Georgia Bight used in this study.** A: Lateral views of the three dentary fragments (GNMH accession numbers, 00-28-09, 00-28-10, 00-28-13), B: Lateral view of the right dentary, (GNMH 4281) with comparison to North Sea specimen (top), Rotterdam Museum of Natural History collections, scale is 20 cm.

(in litt, 19 July 1983), likewise, estimated that this specimen was from a very young calf, probably less than 3 months old. The Hobe Sound skull represents the southernmost record of *E. robustus* in the Atlantic Ocean, based on distributional maps in *Mead & Mitchell (1984)*. Attached to the internal portion of the periotic (Fig. 3) of this juvenile skull is a small sample of sediment, consisting of medium to coarse quartz sand and fragments of mollusk shells, forming a semi-indurated "coquina." This coquina sand, possibly deposited on a beach or in a nearshore marine environment, was probably the original sediment in which the skull was preserved before being dislodged and washed up on the beach.

The second gray whale skull from Florida (FLMNH Specimen UF 99000) was collected during the 1970s by Jesse S. Robertson of Jacksonville University on the beach at Jacksonville Beach, Duval County, northeastern Florida (approximate coordinates, 30°17′ N, 81°23′W) (Fig. 1). Although the collector was unable to remember the specific details surrounding the discovery of this fossil, he did recollect it was found after a strong storm. The Jacksonville Beach gray whale skull consists of a braincase of a juvenile, compared to that of the younger calf from Hobe Sound.

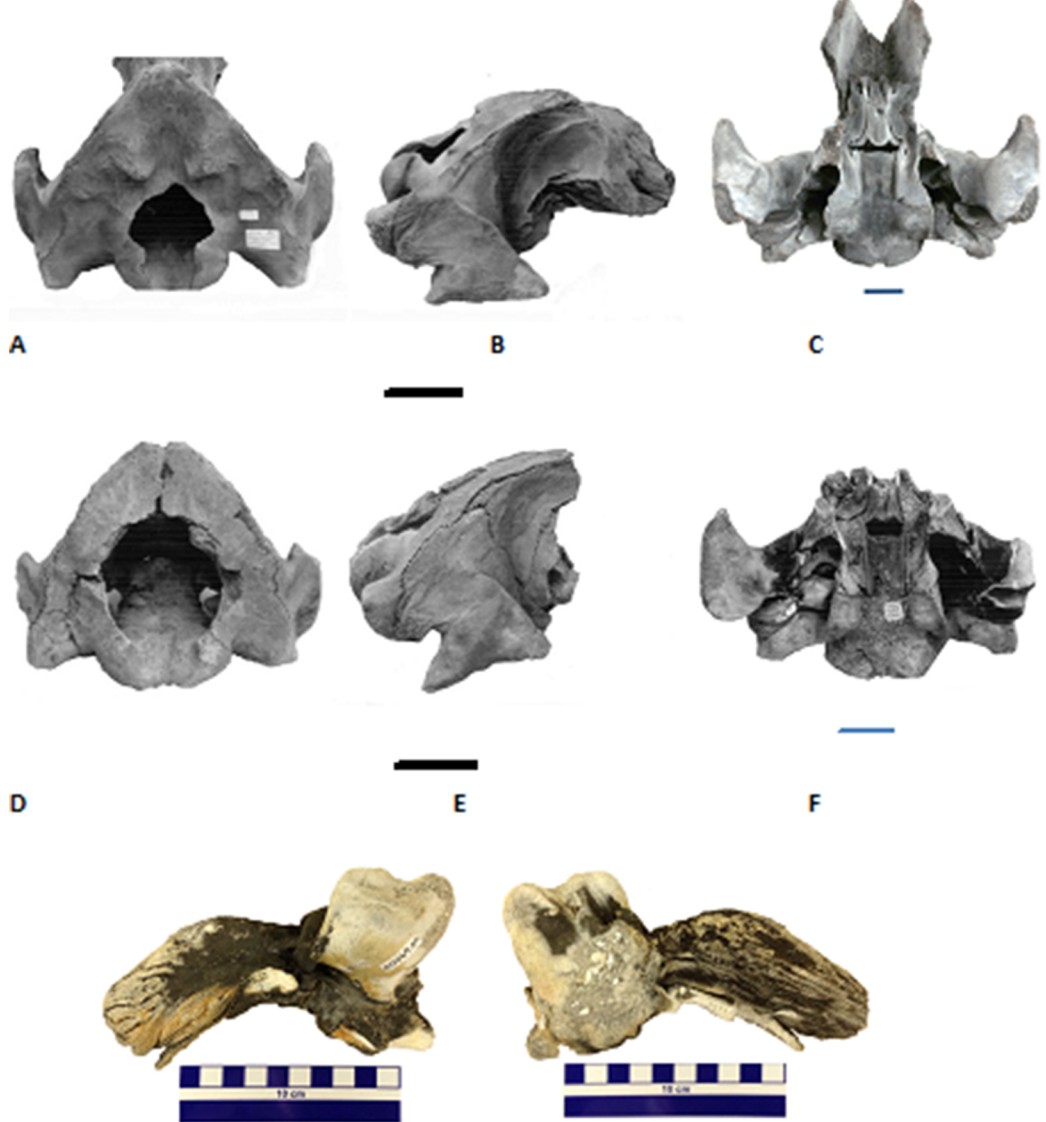

**Figure 3 Two partial Holocene skulls of gray whales (*Eschrichtius robustus*) from Florida (USA).** Top row (A–C), Jacksonville Beach (UF 99000), A. posterodorsal, B. right lateral, and C. ventral views; Middle row (D–F), Hobe Sound National Wildlife Refuge (NWR) (UF 69000), D. posterodorsal, E. right lateral, and F. ventral views; Bottom row, two views of the left periotic (internal ear bone) of the Hobe Sound skull. G. medial and H. lateral views. Both braincases are relatively intact from the condyles to the broken frontals. All scale bars are 10 cm; overall dimensions in Table 1.   

## Carbon isotope analysis and AMS dating

For the stable isotope analysis of samples we used both a Finnigan MAT 252 mass spectrometer, which is a dual-inlet mass spectrometer as well as a National Electrostatics Corporation Model 1.5SDH-1 Pelletron 500 kV compact AMS) unit for precise analyses of carbon isotopes $^{12}$C, $^{13}$C, and $^{14}$C. The Finnigan spectrometer is a double collector gas source mass spectrometer and allows for the measurement of two isotopes of oxygen and three isotopes of carbon with the double collector. It measures the ratio on the sample and the standard by alternating dual inlets. By contrast the Model 1.5SDH-1

Pelletron 500 kV only analyzed for the three isotopes of carbon using Faraday cup collectors at the end of the beam line array.

All samples were quite poorly preserved and did not retain sufficient collagen for AMS analysis, so we applied the bioapatite fraction dating technique (*Cherkinsky, 2009*). Specimens were chemically pretreated to remove all secondary carbonate and organics, but leave bioapatite in the bone structure. Bioapatite belongs to the group of hexagonal calcium phosphates, of which hydroxyapatite, $(Ca_{10}(PO_4)_6(OH)_2$, is closest in structure to biological apatite. It differs from geological apatite by a high degree of isomorphic substitutions and absorption of carbonate and small crystal size, properties that each result in a poorer crystallization of bone bioapatite (*LeGeros & LeGeros, 1984*).

The carbonate substituted within bioapatite maintains its carbon isotope signature in both stable and radioactive isotopes. Carbonate occurs in several locations in the crystals, as absorbed ions on the surface and within the crystals. Substitutions are mainly in the phosphate position and most likely in the hydroxyl position. The absorbed carbonates are more labile but substituted ones are much more stable (they are actually structure carbonates) and contribute to preserving the original isotope composition. Thus, they could be used for radiocarbon dating.

The specimens were treated with a weak acetic acid $(CH_3COOH)$ bath under vacuum. Once washed and dried the samples were treated with phosphoric acid $(H_3PO_4)$ in reaction tubes and heated to ensure removal of all carbonates. The bone samples generated $CO_2$ in the vacuum system and this gas was then analyzed in the mass spectrometer against a Vienna Pee Dee Belemnite standard (VPDB) with the per mil [‰] values of $\delta^{13}C$ reported against VPDB and $\delta^{18}O$ reported against VPDB and Vienna Standard Mean Ocean Water (VSMOW).

We have used the internal standards which are:

| | δ13C, ‰ | δ18O, ‰ |
|---|---|---|
| Fisher | −0.64 | −14.90 |
| A 1296 | +2.56 | −O.60 |

These standards were calibrated with National bureau of Standards (now NIST) NBS 19

δ13C, ‰ = +1.95 and δ18O, ‰ = +28.60.

For AMS isotopic analyses carbon dioxide was cryogenically purified from the other reaction products and catalytically converted to graphite using the method of *Vogel et al. (1984)*. Graphite $^{14}C/^{13}C$ ratios were measured by the Model 1.5SDH-1 Pelletron 0.5 MeV AMS. The sample ratios were compared to the ratio measured from the Oxalic Acid I (NBS SRM 4990). The sample $^{13}C/^{12}C$ ratios were measured separately using a stable isotope ratio mass spectrometer and expressed as $\delta^{13}C$ with respect to VPDB, with an error of less than 0.1‰.

## Collagen peptide mass fingerprinting

We attempted to taxonomically identify five specimens using collagen PMF: the three dentary fragments from JY Reef, Georgia Bight (GMNH 00-28-09, 00-28-10, 00-28-13 and

the two Florida skull specimens (UF 69000, UF 99000). Collagen PMF, also known as ZooMS, is a rapid and cost-effective technique, whereby taxonomic groups are discriminated based on difference in the collagen protein sequence. Robust species identification can be accomplished by comparing collagen peptide fingerprints with the fingerprints from known samples using mass spectrometry (*Collins et al., 2010*). The success of this method has already been demonstrated for ancient North Atlantic cetacean species, including Atlantic gray whale (*Kirby et al., 2013*; *Buckley et al., 2014*).

Sample preparation, mass spectrometry and data analysis followed that described in *Buckley et al. (2014)* and *Evans et al. (2016)* at the BioArCh centre, University of York. Briefly, between 10 and 30 mg of bone powder was fully demineralized through immersion in 0.6M hydrochloric acid, followed by gelatinization in 100 µl of 50 mMol ammonium bicarbonate at 65 °C for 1 h. The resulting collagen was incubated with 0.4 µg of trypsin overnight at 37 °C, acidified to 0.1% trifluoroacetic acid and purified using a 100 µl C18 resin ZipTip® pipette tip (EMD Millipore). The collagen extract was spotted in triplicate on a 384 spot MALDI target plate, with calibration standards and run on a Bruker ultraflex III MALDI TOF/TOF mass spectrometer. mMass software (*Strohalm et al., 2008*) was used to average spectra replicates from each specimen, and compare to the list of m/z markers for marine mammals presented in *Kirby et al. (2013)*, *Buckley et al. (2014)* and *Hufthammer et al., 2018*.

# RESULTS

## Paleontology

If the craniae and dentary elements are that of a gray whale, then systematic paleontology can be:

### SYSTEMATIC PALEONTOLOGY

Class MAMMALIA *Linnaeus, 1758*
Order CETACEA *Brisson, 1762*
Suborder MYSTICETI *Gray, 1864*
Family ESCHRICHTIIDAE *Ellerman & Morrison-Scott, 1951*
Genus ESCHRICHTIUS *Gray, 1864*
ESCHRICHTIUS ROBUSTUS.

*Eschrichtius* cf. *E. robustus* (*Lilljeborg, 1867*).

We compared the two Florida gray whale skulls to descriptions and photographs of modern skulls of *E. robustus* (*True, 1983*, *1904*; *Barnes & McLeod, 1984*) and a late Pliocene skull from Japan referred to *Eschrichtius* sp. (*Ichishima et al., 2006*). Because both Florida skulls consist only of the braincase, our comparisons are limited to characters present in posterior portion of the skull (Fig. 3; Table 1), as well as several characters of the periotics (see next paragraph). Characters the Florida skulls share with *E. robustus* include: triangular-shaped occipital shield with prominent paired occipital tuberosities (see Fig. 3A, UF 99000); large occipital condyles; concave exoccipitals lateral to occipital condyles;

**Table 1 Cranial measurements of Quaternary gray whale skulls.**

| Locality & Catalogue # | Width of squamosals | Width of paraocciptals | Width of supraocciptals | Width of occipital condyles | Height of skull |
|---|---|---|---|---|---|
| Hobe Sound NWR UF 69000 | 60 | 45 | 39 | 22 | 41 |
| Jacksonville Beach UF 99000 | 76 | 59 | 51 | 24 | 51 |

**Note:**
Cranial measurements (in cm) of Quaternary gray whales (*Eschrichtius robustus*) skulls from Florida (this study).

large and posteriorly oriented paroccipital processes; short, massive postglenoid processes of the squamosals that are parabolic in shape; and presence of a squamosal cleft.

Figures 3G and 3H are external (= medial = tympanic) and internal (= lateral = cerebral) views, respectively, of the left periotic from the juvenile gray whale skull from Hobe Sound, Florida (UF 69000). The right periotic of this specimen is still preserved intact in the skull (Fig. 3F). The posterior process of the right periotic (= posterior petrotympanic process of *Ichishima et al., 2006*) is still attached to the skull in the second Florida fossil, from Jacksonville Beach (UF 99000; Fig. 3C). The periotics from the Hobe Sound skull are very similar to periotics from modern skulls of *E. robustus* and to the periotic of a Pliocene skull of *Eschrichtius* from Japan (*Ichishima et al., 2006*). Specifically, the periotic of UF 69000 is similar to that of *E. robustus* in the short, triangular-shaped anterior process, comparatively large cochlear portion (as large as the body and anterior process of the periotic but excluding the posterior process), relatively fibrous posterior process, and the confluence of the fenestra cochleae and the aperture for the cochlear aqueduct. The anterior process of the perotic is proportionally small relative to the cochlear portion (pars cochlearis) in the fossil and *Eschrichtius*, compared to balaenopterids in which the anterior process is much larger.

On the basis of the similarities in the cranial morphology, periotics and ZooMS results, we confidently refer the two Florida skulls to *E. robustus*. Both Florida gray whale skulls represent immature individuals based on the unfused basioccipital/basisphenoid joint or suture. *Walsh & Berta (2011)* examined the closure of the bones in the occipital region in 20 skulls of *E. robustus*, including 18 calves under 1 year of age. They found that the basioccipital/basisphenoid joint becomes completely ossified between 7 months and 1 year of age. The cranial measurements (Table 1), together with the unfused basioccipital/basisphenoid joint, indicate that the Hobe Sound skull (UF 69000) is most likely a newborn calf, whereas, the somewhat larger Jacksonville Beach skull (UF 99000) was probably between 7 months and a year old. Both Holocene skulls of gray whales from Florida are juveniles less than a year of age, lending credence to our suggestion that the east coast of Florida may have been a breeding ground for the western Atlantic population of *E. robustus* (see below under Discussion).

Comparisons with previously described and measured fossils skulls of *Eschrichtius*, as well as modern skulls of *E. robustus*, provide further clarification on the age at death of the two Florida Holocene gray whales. A juvenile fossil gray whale cranium from Cape Lookout, North Carolina measured 80 cm in width across the zygomatic processes (*Mead & Mitchell, 1984*).

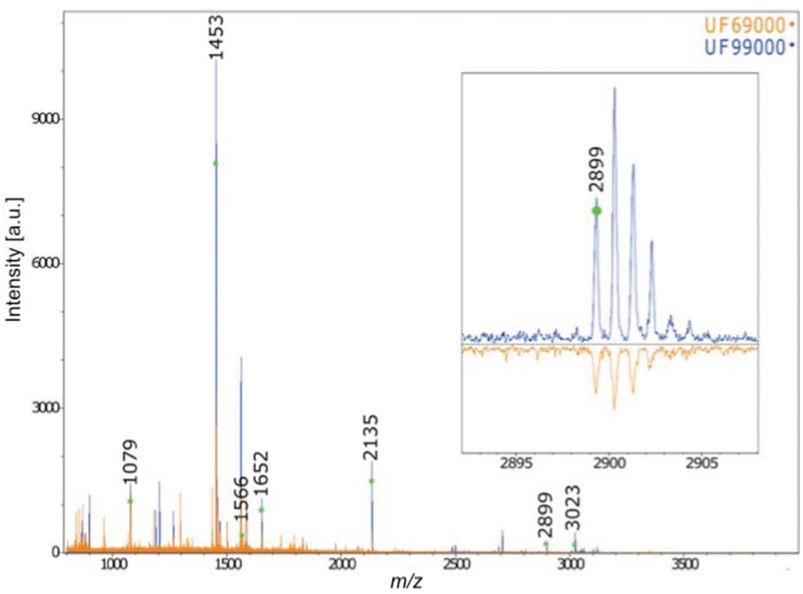

**Figure 4 MALDI-ToF Collagen peptide mass fingerprints for specimens UF 69000 and UF 99000, labeled according to *Buckley et al. (2014)* and *Kirby et al. (2013)*.**

This compares to a width across the zygomatic processes in the two juvenile skulls from Florida of 76 cm in UF 99000 and 60 cm in UF 69000 (width of squamosals in Table 1). An adult fossil gray whale cranium from Corolla, North Carolina measured between 130 and 140 cm across the zygomatics (*Mead & Mitchell, 1984*). The width of the occipital condyles in the adult from Corolla is 31 cm, compared to 24 cm in UF 99000 and 22 cm in UF 69000 (Table 1). A juvenile Pliocene gray whale skull (*Eschrichtius* sp.) from Japan measured 54 cm across the paraoccipital processes (*Ichishima et al., 2006*), compared to widths across the paraoccipitals of 59 cm (UF 99000) and 45 cm (UF 69000) in the two Florida juvenile skulls (Table 1). *Ichishima et al. (2006)* noted that the paraoccipital measurement of the Japanese Pliocene fossil was comparable with the same measurement in a modern gray whale skull from an individual 7.25 m in body length and an age of approximately 30 weeks (*Sumich, Goff & Perryman, 2001*). Comparative data from modern gray whales, including both cranial measurements (*Sumich, Goff & Perryman, 2001*) and fusion of cranial sutures (*Walsh & Berta, 2011*), support our assertion that the Hobe Sound skull was from a calf, younger than 30 weeks based on comparative measurements of the paraoccipitals, whereas, the larger Jacksonville Beach skull was from an animal older than 30 weeks but less than a year old.

Our assessment together with earlier studies, are in line with that of two recently reported Holocene-to-Pleistocene aged gray whale finds in the western Pacific, off Taiwan (*Tsai et al., 2014*) and Japan (*Kimura, Hasegawa & Kohno, 2018*). The former study identified the paired tuberosities on the occipital shield as diagnostic of Eschrichtiidae sp. (*supra*). Both Florida gray whale skulls demonstrate these paired tuberosities, in particular UF 99000 (Fig. 3).

Furthermore, the collagen PMF results confirm the anatomical identification of Gray Whale for the two Holocene-aged Florida skulls (Fig. 4). The two skulls exhibited diagnostic gray whale specific markers presented in *Kirby et al. (2013)* and *Hufthammer et al. (2018)*

**Table 2 Isotopic results for Georgia Bight secimens.**

| Laboratory # | δ¹³C (‰) vs. VPDB | δ¹⁸O (‰) vs. VPDB | δ¹⁸O (‰) vs. VSMOW |
|---|---|---|---|
| Dentary A (00-28-09) | −5.11 | 0.22 | 31.14 |
| Dentary B (00-28-10) | −5.04 | 0.14 | 31.05 |
| Dentary C (00-28-13) | −4.80 | 0.92 | 31.86 |
| 4,281 (dentary) | −6.62 | −10.34 | 20.25 |
| UF 69000 (cranium) | −9.5 | – | – |
| UF 99000 (cranium) | −10.3 | – | – |

Note:
Isotopic results for the Georgia Bight and Florida specimens of *Eschrichtius robustus*.

**Table 3 Radiocarbon ages for specimens.**

| UGA # | Element | Condition | Find location | ¹⁴C age, years BP | Reservoir effect, ΔR | Calendar age, BP 95.4% probability |
|---|---|---|---|---|---|---|
| 4281 | Dentary | fossil | J-Reef | 36,570 ± 300 | −120 ± 78 | 40,230–41,550 |
| 4214 | Dentary? | fossil | J-reef | 33,520 ± 160 | −120 ± 78 | 36,240–37,460 |
| UF 69000 | Cranium | fossil | Jacksonville Beach, Florida | 2,130 ± 25 | −90 ± 131 | 1,500–2,150 |
| UF 99000 | Cranium | fossil | Hobe Sound, Florida | 2,190 ± 20 | −90 ± 131 | 1,570–2,220 |

Note:
Radiocarbon ages for Georgia Bight and Florida specimens of *Eschrichtius robustus*.

(specifically, the presence of diagnostic collagen peptide F at 2899). The application of this technique to our other three Pleistocene-aged specimens, however, failed due a lack of preserved collagen in these ancient bones—a result mirrored in the lack of organic fraction available for radiocarbon dating (see below, and also *Harvey et al., 2016*).

## Radiocarbon (¹⁴C) dates

We dated the samples using the organic fraction and mineral fraction of bioapatite. The overall morphology of the bones was very well preserved, however, the collagen fraction was almost completely destroyed and the concentration of organic carbon was about 0.1% or lower. In all four cases the bioapatite fraction radiocarbon age estimate was older (33–37.5 ka) than that from the organic, collagen-like, fractions (8.3–23 ka). The per mil (‰) values of δ¹³C reported agianst VPDB and δ¹⁸O reported against VPDB and VSMOW and are shown in Table 2, and the AMS dates are displayed in Table 3.

Without doing a full statistical analysis, the results shown in Table 2 for the δ¹³C of sample A (−5.11 ± 0.04‰) and sample B (−5.04 ± 0.04‰) seem to be more strongly related and are probably from the same whale, while sample C (−4.80 ± 0.04‰) differs slightly from the other samples. Sample C could also be from another individual, but because the difference from the other samples is not more than 1‰ it is likely that all three of the mysticete whale bone fragments are from the same individual.

## DISCUSSION

Our results have identified the presence of gray whale in the Georgia Bight and along the Atlantic Coast of Florida, in both the Pleistocene and late Holocene. We are confident
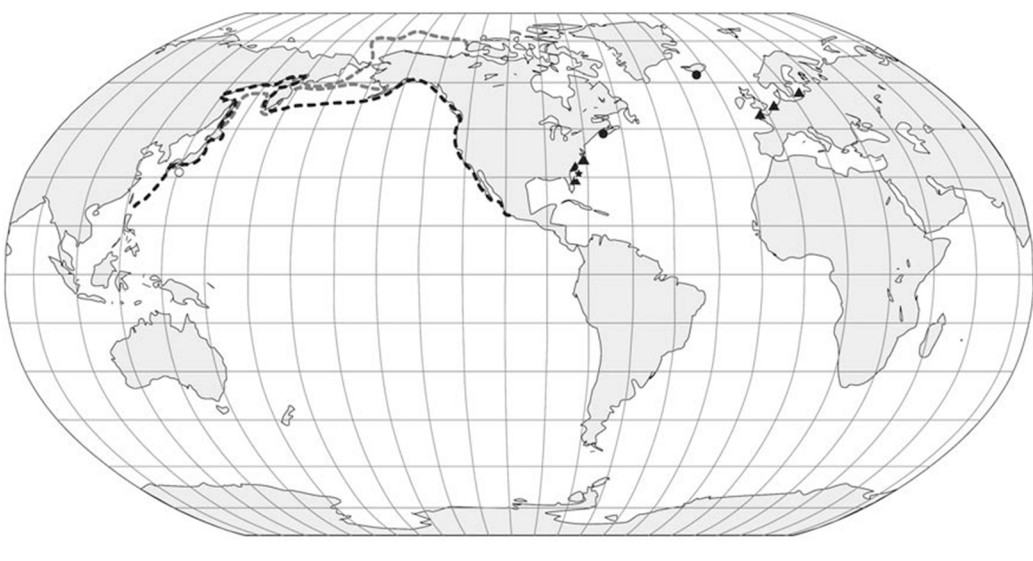

<br>

▲    Fossil evidence for gray whales

●    Historical evidence for gray whales

★    Fossil evidence for gray whales, this study

▢    Summer feeding grounds

-----    Gray Whale migration route

**Figure 5 Global distribution of gray whale fossil finds.**

in the identification of the two craniae, based on both our evaluation and our biomolecular findings using ZooMS and the dentary based on its previous diagnosis (*Garrison et al., 2012*) as this unique mysticete whale; nevertheless, both secure morphological and biomolecular attributes are lacking for the remainder of our sample—three dentary fragments. To include these latter specimens in our assessment of an extirpated population is problematic. We recognize this fact. In the absence of preserved biomolecules, the identity of these remaining specimens has to rest entirely on morphological identification criteria and provenance. In the case of the latter, this is to say that their find locations were in close proximity to UGA 4281 and later discovered cranial elements were identified as *Eschrichtius* in previous study (*Noakes, Pyenson & McFall, 2013*). By parsimony, we argue that if these are whale then they most likely are gray whale.

These observations, in particular the two late Holocene records of juveniles from Florida, potentially provide insight into this species' migration route and calving grounds. The eastern Pacific population of gray whales annually migrates from summer grounds in the Bering Sea and Chukchi Sea between Alaska and Russia to winter/calving grounds in Mexico along the western coast of Baja California and the southern Gulf of California (Fig. 5). The best known calving sites of *E. robustus* in the eastern Pacific are located in shallow lagoons along the coast of Baja California between 24° and 29° North latitude, including Laguna Ojo de Liebre (also known as "Scammon's Lagoon"),

**Table 4 Pleistocene and Holocene Atlantic gray whale specimens.**

| Region | Date found | Cal yr BP | Citation | Current Location |
|---|---|---|---|---|
| Gray whale—Eastern Atlantic Basin | | | | |
| Pentuan, England | 1829 | 1,329 ± 195 | *Flower (1872)*; *Bryant (1995)*; *Alter et al. (2015)* | unknown |
| Gräsö, Sweden | 1859 | 4,395 ± 155 | *Lilljeborg (1861)*; *Persson (1986)* | unknown |
| Babbacombe Bay, England | 1861 | | *Gray (1864)* | unknown |
| Babbacombe Bay, England | 1865 | 340 ± 260 | *Gray (1866)*; *Pengelly (1865, 1878)*; *Bryant (1995)*; *Alter et al., 2015* | unknown |
| Ijmuiden, Netherlands | 1879 | 8,330 ± 85 | *Van Deinse & Junge (1937)*; *Bryant (1995)* | National Natural History Museum Naturalis, Leiden |
| Ijmuiden, Netherlands | 1916 | 1,400 | *Van Deinse & Junge (1937)*; *Bryant (1995)* | National Natural History Museum Naturalis, Leiden |
| Wieringermeer-Polder, Netherlands | 1935 | 4,195 ± 45 | *Van Deinse & Junge (1937)*; *Bryant (1995)* | National Natural History Museum Naturalis, Leiden |
| Oostduinkerke-Koksijde, Belgium | 1978 | 2,024 ± 110 | *Asselberg (1981)*; *Bryant (1995)* | Unknown |
| North Sea, Netherlands | 2005 | 42,800 ± 4,100–2,700 | *Post & Bosselaers (2005)* | Natural History Museum, Rotterdam |
| North Sea, Netherlands | 2005 | >45,200 | *Mol et al. (2006)* | Natural History Museum, Rotterdam |
| North Sea, Netherlands | 2001 | 1,150–1,270 | *Alter et al. (2015)* | Natural History Museum, Rotterdam |
| North Sea, Netherlands | No Data | 1,350–1,500 | *Alter et al. (2015)* | Unknown |
| North Sea, Netherlands | 1997 | 1,350–1,500 | *Alter et al. (2015)* | Natural History Museum, Rotterdam |
| North Sea, Netherlands | 2003 | 2,650–2,730 | *Alter et al. (2015)* | Natural History Museum, Rotterdam |
| North Sea, Netherlands | No data | >48,000 | *Alter et al. (2015)* | Unknown |
| North Sea, Netherlands | 2005 | >48,000 | *Alter et al. (2015)* | Natural History Museum, Rotterdam |
| North Sea, Netherlands | 2003 | 42,500–43,300 | *Alter et al. (2015)* | Natural History Museum, Rotterdam |
| North Sea, Netherlands | 1879 | 9,470–9,550 | *Alter et al. (2015)* | National Natural History Museum Naturalis, Leiden |
| North Sea, Netherlands | 1916 | 1,600–1,800 | *Alter et al. (2015)* | National Natural History Museum Naturalis, Leiden |
| North Sea, Netherlands | 1935 | 4,760–4,850 | *Alter et al. (2015)* | National Natural History Museum Naturalis, Leiden |
| North Sea, Netherlands | No data | 4,950–5,250 | *Alter et al. (2015)* | Natural History Museum, Rotterdam |
| North Sea, Netherlands | 1954 | 3,830–3,960 | *Alter et al. (2015)* | Natural History Museum, Rotterdam |
| North Sea, Netherlands | 1994 | 960–1,120 | *Alter et al. (2015)* | Natural History Museum, Rotterdam |
| North Sea, Netherlands | 1995 | 4,230–4,420 | *Alter et al. (2015)* | Natural History Museum, Rotterdam |
| North Sea, Netherlands | 1996 | >48,000 | *Alter et al. (2015)* | Natural History Museum, Rotterdam |
| North Sea, Netherlands | 2005 | 1,820–1,950 | *Alter et al. (2015)* | Natural History Museum, Rotterdam |
| North Sea, Netherlands | 2005 | >50,000 | *Alter et al. (2015)* | Natural History Museum, Rotterdam |
| North Sea, Netherlands | 2005 | 3,480–3,630 | *Alter et al. (2015)* | Natural History Museum, Rotterdam |
| North Sea, Netherlands | 2005 | 10,000–10,180 | *Alter et al. (2015)* | Natural History Museum, Rotterdam |
| North Sea, Netherlands | 2005 | 5,280–5,430 | *Alter et al. (2015)* | Natural History Museum, Rotterdam |
| North Sea, Netherlands | 2005 | 6,620–6,700 | *Alter et al. (2015)* | Natural History Museum, Rotterdam |
| North Sea, Netherlands | No data | 5,320–5,470 | *Alter et al. (2015)* | Unknown |
| North Sea, Netherlands | No data | 3,470–3,620 | *Alter et al. (2015)* | Unknown |

(Continued)

 

| Region | Date found | Cal yr BP | Citation | Current Location |
|---|---|---|---|---|
| North Sea, Netherlands | No data | 40,200–41,400 | *Alter et al. (2015)* | Natural History Museum, Rotterdam |
| North Sea, Netherlands | No data | 1,680–1,800 | *Alter et al. (2015)* | Natural History Museum, Rotterdam |
| North Sea, Netherlands | No data | 42,400–43,600 | *Alter et al. (2015)* | Natural History Museum, Rotterdam |
| North Sea, Netherlands | 2007 | 3,930–4,070 | *Alter et al. (2015)* | Natural History Museum, Rotterdam |
| North Sea, Netherlands | No data | 4,020–4,270 | *Alter et al. (2015)* | Unknown |
| Gray whale—Western Atlantic Basin | | | | |
| Tom's River, New Jersey | 1850s | 455 ± 90 | *Mead & Mitchell (1984)* | Smithsonian National Museum of Natural History, Washington D.C |
| Myrtle Beach, South Carolina | 1959 | 865 ± 165 | *Mead & Mitchell (1984)* | unknown |
| Chesapeake Bay, Virginia | 1969 | 10,140 ± 125 | *Mead & Mitchell (1984)* | unknown |
| Nags Head, North Carolina | 1970's | 865 ± 50 | *Mead & Mitchell (1984)* | Smithsonian National Museum of Natural History, Washington D.C |
| Corolla, North Carolina | 1976 | 2,415 ± 90 | *Mead & Mitchell (1984)* | unknown |
| Southampton, New York | 1977 | 275 ± 35 | *Mead & Mitchell (1984)* | unknown |
| Corolla, North Carolina | 1977 | | *Mead & Mitchell (1984)* | unknown |
| Rehobeth, Delaware | 1978 | | *Mead & Mitchell (1984)* | Smithsonian National Museum of Natural History, Washington D.C |
| Cape Lookout, North Carolina | 1979 | 1,190 ± 245 | *Mead & Mitchell (1984)* | unknown |
| Jupiter Island, Florida | 1983s | ~1,500–2,150 | this study | Florida Museum of Natural History (UF69000) |
| Jacksonville Beach, Florida | 1970s | 1,570–2,220 | this study | Florida Museum of Natural History (UF99000) |
| South Atlantic Bight, Georgia | 2006 | ~36,000 yBP | *Noakes, Garrison & McFall (2009)*; *Cherkinsky, 2009*; *Garrison et al. (2012)* this study | Georgia Museum of Natural History (No. 4032) |
| South Atlantic Bight, Georgia | 2006 | 41,490–42,070 | This study | Georgia Museum of Natural History (No. 4024) |
| South Atlantic Bight, Georgia | 2006 | 40,230–41,550 | This study | Georgia Museum of Natural History No. 4281) |
| South Atlantic Bight, Georgia | 2006 | 38,350–39,140 | This study | Georgia Museum of Natural History (No. 4215) |
| South Atlantic Bight, Georgia | 2006 | 36,240–37,460 | This study | Georgia Museum of Natural History (No. 4214) |
| South Atlantic Bight, Georgia | 2006 | 48,550–50,000 | This study | Georgia Museum of Natural History (No. 7742a) |

**Note:**
  Pleistocene and Holocene eastern and western Atlantic gray whale specimens.

Laguna San Ignacio, and Bahia Magdalena (*Rice & Wolman, 1971*). The presence of a fossil skull (UF 69000) representing a newborn gray whale calf from Jupiter Island along the southeastern coast of Florida suggests the possibility that this region may have been used as a calving ground by the now-extinct western Atlantic population of gray whales. Along the Atlantic coast of southeastern Florida between 24° and 29° N there are numerous shallow bays and protected lagoons, including Florida Bay, Biscayne Bay, Lake
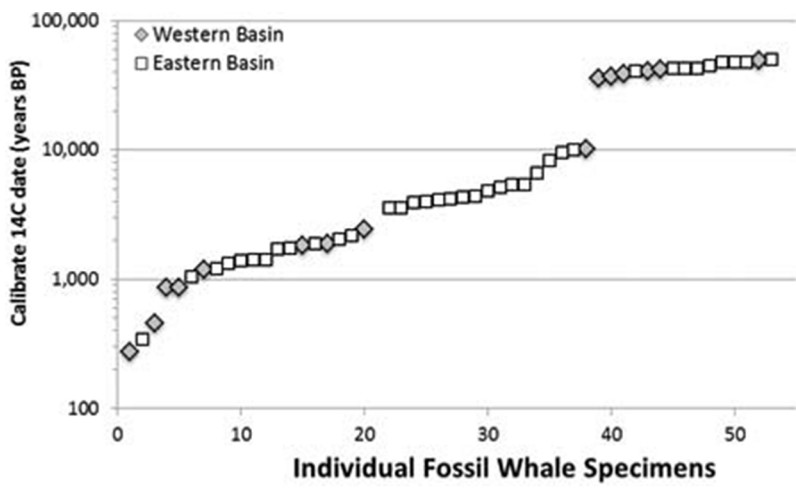

**Figure 6 Scatter plot of 53 radiocarbon ages for Atlantic Basin gray whale finds, with gap between ~35,000 and ~11,000 BC (Logarithmic scale, base 10).**

Worth, Hobe Sound, Indian River, Banana River, and Mosquito Lagoon, that would seem to have provided ideal calving grounds for gray whales.

Our results also provide additional insights into the ecological history of this enigmatic species. Current Atlantic Arctic species have evolved over periods whereby adaptation to profoundly different climate regimes was required, such as ca. 12,000 years ago when the Bering land bridge closed the western Arctic to Pacific water intrusion (*Walsh, 2008*) or during the retreat of the great ice sheets and the opening continental shelves at the onset of the Holocene (*Harington, 2008*). Climatic changes of the glacial cycles are thought to have been a major driver of arctic population declines and species extinctions, however, there is still not a full understanding of how marine species responded to past climate change.

It has been suggested that some Pleistocene cetacean lineages survived into the Holocene and their effective female population size increased rapidly, concurrent with a threefold increase in core suitable habitat (*Evans, 1987*). For example, using ancient DNA analysis, *Foote et al. (2013)* show that the bowhead whale (*Balaena mysticetus*), shifted its range and tracked its core suitable habitat northwards during the rapid climate change of the Pleistocene–Holocene transition. The case for this type of habitat tracking in Late Pleistocene Atlantic gray whale is not as straightforward. *Pyenson & Lindberg (2011)* argue for the adaptability of gray whales, suggesting that gray whales survived Pleistocene glacial maxima (e.g., LGM) and maintained substantial population sizes by employing a diverse set of feeding modes, similar to those seen in seasonal resident whales found today between northern Washington State and the coast of Vancouver Island. Molecular data, however, do not support a widespread (maternal) continuity in gray whale lineages across the LGM. In their analysis of Atlantic gray whale fossils, *Alter et al. (2015)* detected little genetic continuity between the late Pleistocene and Holocene populations—within the dataset, only a single sample displayed a lineage that survived post-LGM. The remaining Holocene samples from both the

Western and Eastern basins shared a most recent common ancestor with Pacific gray whales dating to the early to mid-Holocene, suggesting that the majority of the Atlantic Holocene population were the result of a second colonization event when warming temperatures, sea-level rise, and decreases in sea ice permitted passage through the Bering Strait (*Alter et al., 2015*).

We examined these two hypotheses in light of the range and distribution of the 53 radiocarbon dates for all available Atlantic gray whale finds (Fig. 5). There are 15 late Pleistocene ages that range from >45,200 to 35,520 BP and 38 Holocene ages that range from 10,400 to 340 BP (Table 4; Fig. 6). These ages are clearly bimodal in their distribution suggesting either that: (1) any population surviving post-LGM is geologically "invisible," due to a lack of fossil evidence; or (2) the lack of finds reflects their true absence, indicating a significant decline or even the effective extirpation of gray whales across the north Atlantic Ocean in the period between ~40 and ~11 ka. The argument in favor of the latter may be further supported by geologic evidence for a much-reduced habitat in the north Atlantic during LGM due to the subaerial exposure of both the North Sea/Baltic and the Georgia Bight and Florida continental shelves (*Alter et al., 2015*; *Garrison, McFall & Noakes, 2008*; *Garrison et al., 2012*; *Harris et al., 2013*). Fossil evidence for the gray whale, however, may yet be found for the post- 40–11 ka interval on areas of the Atlantic continental shelf that were inundated at or during the last low stand. Nevertheless, the dates produced in our study for recent western Atlantic Ocean finds are in good alignment with these (predominantly Eastern basin) dates and, likewise, suggest a bi-modal distribution for Atlantic gray whales in the Western basin.

## CONCLUSIONS

In our results, collagen PMF and paleontological diagnoses confirm the two Florida finds to be the Atlantic gray whale, *E. robustus*. Due to the lack of preserved collagen in the three Pleistocene aged fragmentary specimens, we cannot confirm the species identities through biomolecular analyses; paleontological diagnoses, however, offer support for our identification of the relatively well-preserved Georgia Bight dentary as gray whale as well as findings based on our previous diagnosis (*Garrison et al., 2012*). Thus, the existence of both Pleistocene and Holocene gray whale populations in the western Atlantic Ocean are supported by these fossils and their radiocarbon dates. The results of our study may help explain the Late Quaternary occurrences of gray whales in the Georgia Bight and along the Atlantic Coast of Florida, in particular the two late Holocene records of juveniles from Florida. The presence of a newborn gray whale calf's skull from Jupiter Island as well as a juvenile gray whale's dentary from the Georgia Bight suggests that the region between 24° and 29° N may have been used as a calving ground by the now-extinct western Atlantic population of gray whales.

## ACKNOWLEDGEMENTS

We thank the following colleagues and/or agencies for their assistance in the research that contributed to this article:

Drs. Brenna McLeod and Tim Frazier, St. Mary's University, Halifax, Nova Scotia, Canada;

Dr. Michael Hofreiter, Potsdam University, Germany;

Klaas Post, Natural History Museum, Rotterdam, The Netherlands;

Keri Rowsell for technical assistance, University of York, UK;

C.W. (Kees) Moeliker & H.P. (Henry) van der Es, Natural History Museum, Rotterdam, The Netherlands;

Wendy van Bohemen, Naturalis Biodiversity Center, Leiden, The Netherlands;

Dr. Scott E. Noakes, Center for Applied Isotope Studies, University of Georgia, Athens, GA;

Dr. Mark Williams; Amanda Thompson; Isabelle Cantin—The Laboratory for Archaeology, University of Georgia, Athens, GA;

Dr. Nicholas Pyenson, Smithsonian Institution, Washington, DC; and the National Oceanic and Atmospheric Administration (NOAA).

### Funding

Funding for molecular identification was provided through ORCA FP7-PEOPLE-2011-IOF 299075. The funders had no role in study design, data collection and analysis, decision to publish, or preparation of the manuscript.

### Grant Disclosure

The following grant information was disclosed by the authors:
Molecular identification was provided through: ORCA FP7-PEOPLE-2011-IOF 299075.

### Competing Interests

The authors declare that they have no competing interests.

### Author Contributions

- Ervan G. Garrison conceived and designed the experiments, performed the experiments, analyzed the data, contributed reagents/materials/analysis tools, prepared figures and/or tables, authored or reviewed drafts of the paper, approved the final draft.
- Gary S. Morgan conceived and designed the experiments, performed the experiments, analyzed the data, contributed reagents/materials/analysis tools, prepared figures and/or tables, authored or reviewed drafts of the paper, approved the final draft.
- Krista McGrath conceived and designed the experiments, performed the experiments, analyzed the data, contributed reagents/materials/analysis tools, authored or reviewed drafts of the paper, approved the final draft.
- Camilla Speller conceived and designed the experiments, performed the experiments, analyzed the data, contributed reagents/materials/analysis tools, prepared figures and/or tables, authored or reviewed drafts of the paper, approved the final draft.

- Alexander Cherkinsky conceived and designed the experiments, performed the experiments, analyzed the data, contributed reagents/materials/analysis tools, authored or reviewed drafts of the paper, AMS radiocarbon dating.

## Data Availability

Morphometric measurements are in the tables and all specimens are clearly shown with scales in Figs. 2 and 3. The isotopic and peptide data are shown in tables and Figures (Fig. 4). All data are archived and available at the laboratories that performed the analyses. AMS-radiocarbon data for the published specimen ages are archived at the Center for Applied Isotopic Studies (CAIS), University of Georgia, USA; the ZooMS data are archived with the BioArCh Laboratory, Department of Archaeology, York University, UK; all specimens utilized for the study are curated at either: (1) Georgia Museum of Natural History, University of Georgia; or (2) Florida Museum of Natural History, University of Florida, USA. The dentary fossil shown in Fig. 2, for comparative purposes, is curated at the Rotterdam Museum of Natural History, Rotterdam, Netherlands. No numeric data were used for that specimen.

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
