# Peer review of "Recent dating of extinct Atlantic gray whale fossils, (Eschrichtius robustus), Georgia Bight and Florida, western Atlantic Ocean"

_PeerJ, doi:10.7717/peerj.6381_

## Round 0.1 · original submission · Major Revisions

I received assessments from three reviewers. In general, everyone thinks your publication is a good contribution, and that fits well with PeerJ goals. However, they have also pointed out that several problems in your paper that require a substantial revision.

All reviewers provided valuable feedback. You should follow all of them when reviewing your paper. Besides, one of the reviewers pointed out that the “publication of specimens that are not properly curated and accessioned in institutions that constitute a public trust is strictly forbidden under ethical guidelines of the Society of Vertebrate Paleontology.” I understand that specimens are under permanent loan to an academic institution, but that is not enough. One of the reviewers suggested that “at the very least I encourage the authors to more fully address the specifics of this "permanent loan."

When resubmitting your manuscript, please carefully consider ALL points mentioned in the reviewers' comments, explain every change made, and provide proper rebuttals for any remarks not addressed.

Reviewer 1 ·

Basic reporting

The MS is written in clear and unambiguous English. A few typos are noted throughout the attached annotated PDF, but otherwise the prose is quite good. The literature references are up to date and thorough. One suggested addition is Peredo and Uhen 2016 (Palaeogeography, Palaeoclimatology, Palaeoecology), which discusses marine mammal biogeography via the arctic passageways. The article, figures, and tables, are all structured according to house style. The MS is written to be concise and to restrict itself to the relevant hypotheses and data with no off topic speculation.

Experimental design

The MS falls entirely within the aims and scope of the journal and is original, primary research of good quality. The questions are well defined, the data address their questions, and their conclusions are reasonable given the results. All experiments are performed to high standards, and the methods are sufficiently described to facilitate replication.

Validity of the findings

The data support the overall findings in such a way that I see little reason to disagree with the interpretation presented by the authors. I do not doubt the ages of the fossils present, nor the identity of the fossils presented. The authors are modest in their phrasing, suggesting that these fossils may suggest a calving ground: an assertion that I think is reasonable given the data presented.

Annotated reviews are not available for download in order to protect the identity of reviewers who chose to remain anonymous.

Reviewer 2 ·

Basic reporting

The manuscript describes a number of fossil finds assigned to the gray whale Eschrichtius robustus from the western North Atlantic ocean and investigates into their ages through Accelerator Mass Spectrometry (AMS). As a whole, the manuscript is well written with good technical English and does not require further revision. The research included in this manuscript is interesting for a technical, international audience. As the results are interesting and well grounded, the manuscript is worth publishing but only after minor revision.

English: good.
Literature: good.
Article structure: minor revision
Results: good.

Experimental design

The experimental design seems appropriately realized. Techniques and methods are adequately explained and only minor changes are necessary in order to increase the readability of the manuscript for a broader audience.

Original primary research within aims and scope of the journal.
Research question well defined, relevant and meaningful. It is stated how research fills an identified knowledge gap.
Rigorous investigation performed to a high technical and ethical standard.
Methods description needs only a minor revision to be sufficiently well described.

Validity of the findings

The findings are globally and methodologically valid. The chronological placements of these fossil whales provide new and interesting data to understand the evolution of the gray whale populations in the North Atlantic in the Pleistocene and Holocene. Moreover, the hypothesis that sites along the coasts of Florida formed reproductive areas for gray whales in the Pleistocene and Holocene is interesting and useful to understand patterns and processes of extinction in this species.

The manuscript presents novel interpretations and new results for the field. Data is robust, statistically sound and controlled. Conclusion need a minor revision. The speculative part of the manuscript (dealing with proposed reproductive sites for Pleistocene and Holocene gray whales) is well identified as such.

Additional comments

My comments are listed below.
Line 82: AMS: please, provide citations to support that this method is appropriate for the goals of the paper.
Lines 148 and 168: provide geographic coordinates in a consistent manner: you used 27°07’40” North latitude and 80°08’45” West longitude and 30°17’N, 81°23’W; the latter way is the best.

Line 157: Justify the statement that the specimen “is from a very young individual or calf probably less than a year old”.

Lines 140 and 162: use coquina or “coquina” consistently throughout the manuscript.

Line 183: you have parentheses to open somewhere.

Lines 238-241: please explain what are the eschrichtiid attributes of these vertebrae; moreover, explain why the vertebrae are assigned to the thoracic section of the vertebral column because, judging from the image you provided, at least one seems part of the lumbar section because the transverse processes start from approximately the middle of the height of the vertebral body. Additionally, a synthetic description of the periotic is needed.

Lines 249-255: details of the temporal fossa are necessary. What are the relationships of the alisphenoid with the other bones; what is the shape of the squamosal cleft? Additional figures and descriptions are necessary if the authors want to publish a paleontological section of this paper. Otherwise, remove all the paleontological part.

Lines 258-259: I was unable to find any information in Table 1 to confirm the juvenile age of the specimen. Why the authors cited Table 1 in these lines? Please, explain or remove the Table 1 citation.

Line 275: The authors identified the previous paragraph under the title Paleontology. Here a new section is starting that needs a paragraph title. After the title, remove Furthermore, and rephrase accordingly to the new paragraph structure.

Line 373: change e.g. into hereinafter:

Line 378: use the symbol – consistently in this phrase.

Line 392: Correct the blank space.

Lines 384-399: report your finds with more details. You should report the dates of the fossils here before making a synthetic discussion.

Judging from Fig. 5, there is also a gap in the interval between 100 and 1000 14C calibrate date. Can you discuss this gap? Moreover, change 14C into 14C in the label for the calibrate date in the figure.

References
Check the reference list strictly. Below, I list some problems I found but my list is not complete. Be consistent and strictly follow the instructions to realize your reference list. For example: you wrote 51(2):647-655 for Cherkinsky (2009) and 26, 405–413 for de Bruyn et al. (2011). Moreover, citations are sometimes incomplete (Collins et al. 2010 lacks pages; Alter et al. is cited as volume 24.7); check also George et al. (2004) for comma placement, McLeod et al. (2008) lacks the number of the volume etc. etc.

Line 419: include all the authors in the Alter et al. paper.
Lines 440-441: add a blank line separating the citations.
Lines 447-448: remove “ and ” from the citation.

Illustrations
Fig. 2: add inventory number of Rotterdam specimen; B represents a medial view of both specimens not a lateral view. Change the caption. Why parts A and B of Fig. 2 are not together with parts C and D? It seems that they belong to different figures. Authors must provide also lateral and ventral views of these vertebrae to confirm assignments to the thoracic region of the vertebral column; in fact, judging from the provided photographs, at least two vertebrae seem to belong to the lumbar section as the bases of their transverse processes are located approximately at the middle of the height of the vertebral body.
Fig. 3: illustrations are very small and difficult to read. There is a lot of space that can be filled with larger illustrations. I recommend authors to enlarge the single illustrations (views of the skulls and periotics) in order the audience can better understand the morphologies of the specimens.
Table 1: what do exactly the author mean for “paraoccipital”? Additional measurements are needed in order to can include these specimens in broader morphometric datasets. I list these below:
a) Maximum height and maximum width of foramen magnum;
b) Maximum length of supraoccipital from higher border of the foramen magnum to its apex;
c) Maximum width of foramen magnum + width of both occipital condyles (this may be used for size reconstruction of the whale);
d) Bizygomatic width.
There are no measurements of the periotic. The following measurements are needed:
a) Length of posterior process;
b) Maximum width of posterior process;
c) Lateromedial diameter of pars cochlearis;
d) Anteroposterior diameter of pars cochlearis;
e) Maximum height of the periotic;
f) Maximum and minimum diameters of internal acoustic meatus;
g) Maximum and minimum diameters of fenestra ovalis;
h) Maximum and minimum diameters of fenestra rotunda.

Reviewer 3 ·

Basic reporting

See attached file.

Experimental design

See attached file.

Validity of the findings

See attached file.

Additional comments

See attached file.

Annotated reviews are not available for download in order to protect the identity of reviewers who chose to remain anonymous.

---

## Round 0.2 · Major Revisions

Your paper has been reviewed again by the previous reviewers. Two of them are fine with the version, but one of them (Reviewer 2) found that the manuscript still needs several improvements. I think the points raised by the second reviewer are valid and should be followed strictly to make your article ready for publication. I invite you to submit an updated version of your manuscript with all changes proposed by the reviewers.

Reviewer 1 ·

Basic reporting

This study is written in clear and professional English. The references are sufficient and and the manuscript presents all arguments, evidence, and rationale in clear and consistent formats.

Experimental design

No comment.

Validity of the findings

No comment.

Reviewer 2 ·

Basic reporting

The manuscript appears slightly modified from the previous version. Illustrations are better prepared and can be easily observed. Some elements of the text were modified accordingly to the reviewers' recommendations but, as stated by the authors in their rebuttal letter, they maintain some views about their methods that the reviewers criticized. In particular, the paleontological analysis is superficial and certainly insufficient to support the taxonomic decisions made by the authors about the vertebrae. The paleontological section is quite superficial with insufficient descriptions and measurements of skulls, mandible and vertebrae. The vertebrae should be removed from the manuscript otherwise their assignment to Eschrichtiidae has to be better established. If the authors want to maintain the vertebrae in the manuscript then the manuscript needs major revisions before being accepted for publication; if the authors agree to remove the vertebrae then the manuscript needs minor revisions. Certainly, as detailed in the other sections of this review, in its present version, this manuscript should not be accepted for publication as a number of revisions is still to be made by the authors.

Experimental design

Original primary research within Aims and Scope of the journal.

Research question well defined, relevant & meaningful. It is stated how research fills an identified knowledge gap.

The paleontological investigation is insufficient. The methods followed in the paleontological analysis are not sufficiently explained and the authors failed to provide enought details to replicate their observations and to support their taxonomic conclusions about the vertebrae. The methods followed for skull descriptions are not sufficiently explained and the resulting descriptions are too superficial.

Validity of the findings

While the skulls can be assigned to Eschrichtius based on the illustrations, questions still exist about the vertebrae and the dentary. Moreover, the insufficient descriptions of the skulls and periotic do not allow the clear identification of the specimens as Eschrichtius robustus so that a hypothesis of a different species assignment still holds. More detailed descriptions are necessary to better support taxonomic decisions. In the end, the conclusions are not sound for a number of specimen examined in this manuscript.

Additional comments

The paleontological section is quite superficial with insufficient descriptions and measurements of skulls, mandible and vertebrae. In the rebuttal letter, the authors wrote that their assignment of the fossils to Eschrichtius is based firstly on identificiations made by other authors but they failed to document on which bases these authors made their choice. This is somewhat an Aristotelian way to go: as these authors said that the bones belong to Eschrichtius then these bones belong to Eschrichtius. This reasoning is not acceptable in a scientific manuscript. The readers have the right to know and the authors of the current manuscript have the duty to explain what characters were used to assign the vertebrae to Eschrichtius in order enable other students to do the same observations and eventually falsify their hypothesis. Presently, their assignment of the vertebrae to Eschrichtius is based on a comparison between morphometric characters that is not fully explained (see below). Frankly speaking, I would remove the vertebrae from the manuscript or state that they are belonging to Mysticeti gen. et sp. Indet. Even in this case, the skulls and the dentary are enough to support the paleoecological hypothesis of the authors about the breeding ground.
I must stress what written by the authors in the rebuttal letters as they stated that the requested additional descriptions and measurements are not necessary because out of the scope of the paper. In fact, the authors stated that in their opinion these requests would be useful for taxonomic purposes only but their goals are different. They were wrong: additional descriptions and measurements are needed to better describe the specimens and to support the taxonomic conclusions. From the present manuscript, taxonomic conclusions are really scarcely supported, therefore a full description of the specimens would provide a better support: they need to have more characters to support their taxonomic decisions. I strongly recommend to add morphological details to the paleontological section of this manuscript to better support both the taxonomic decision and the paleoecological hypothesis.
Line 47: A monotypic taxon is a group (taxon) including only one immediately subordinate taxon. Based on this manuscript, this means that Eschrichtiidae should include only Eschrichtius. This is not the case: there are four putative genera of Eschrichtiidae: Eschrichtius, Archaeschrichtius, Eschrichtioides and Gricetoides. Thus, the family is not monotypic. The authors have to state, eventually, that the family Eschrichtiidae includes only one living genus and species, Eschrichtius robustus.
Line 47: it is interesting to note that the authors include only one morphology-based citation about the phylogenetic and taxonomic relationships of Eschrichtiidae (i.e., McLeod et al. 1993). It would be useful to cite also more recent works of this kind such as Deméré et al. (2005; Journal of Mammalian Evolution), Steeman 2007; Zoological Journal of the Linnean Society), Bisconti (2008; Zoological Journal of the Linnean Society), Whitmore & Kaltenbach (2008; Virginia Museum of Natural History Special Publication).
Impossible to open fig. 4
Line 258: in my knowledge, the remnants of the lateral processes are not diagnostic at family level for Eschrichtiidae. If the authors think that this character is diagnostic than they have to demonstrate that. If they are unable to perform such a demonstration, they have to remove this character from the list.
Line 258: overall centrum size: this parameter is linked to the ontogenetic age of the individuals. Are the authors argumenting that the overall centrum size of these juvenile vertebrae are consistent with values for this parameter as scored from living gray whale individuals of the same ontogenetic ages? Did they compare their measurements with a sample of gray whale individuals of different ages? In general, the overall centrum size is not a good proxy for a morphological identification of a whale species unless the size is really peculiar (e.g., in the adult blue whale).
Line 259: at least balaenopterid in origin: this means that there is no certainty about the assignment of these vertebrae to Eschrichtiidae as there is a possibility that they belong to a balaenopterid whale. The paleoecological conclusions of the authors depend upon the clear and correct identification of these fossils as belonging to gray whales; if this identification is not possible beyond each reasonable doubt then the conclusions are weakened. This phrase undermines a strong uncertainty in this research and makes really weak conclusions.
Line 259: it is not correct to talk about “key data” as the datum is only one: the ratio of centrum length to centrum height. The authors have to state that their assignment of this vertebra to Eschrichtiidae depends upon this measurement only.
Line 266: In general, there are quite clear criteria to distinguish cervical, thoracic, lumbar and caudal vertebrae in a mysticete skeleton. Thoracic vertebrae have the transverse processes located in the anterior-most part of the lateral wall of the centrum and bears articular facets for the heads of the ribs. This characters should be sufficient to assign the vertebrae to the thoracic section of the vertebral column. Do the authors can check this character and report in the manuscript about it? Noting the position of the remnants of the lateral (better: transverse) processes would be feasible in the materials they are studying.
Line 291: squamosal cleft is present also in Balaenopteridae and some Cetotheriidae. What is the shape of the squamosal cleft? In Balaenopteridae the squamosal cleft has a triangular shape with a dorsally directed apex; in Cetotheriidae and Eschrichtiidae the shape is straight. Noting the presence of a squamosal cleft is insufficient to make an assignment to Eschrichtiidae, description is necessary.
Impossible to open fig. 6
If the taxonomy was correct, then the paleoecological conclusions would be sound but, unfortunately, the taxonomic decisions made by the authors are biased by a number of issues. I addressed these issues in the line-per-line comments. In the present version of the manuscript, the paleontological section is still superficial and unable to support the taxonomic decisions about the vertebrae. Additional descriptions of the skulls are necessary to allow the readers to fully understand morphology and taxonomy of the specimens.

Reviewer 3 ·

Basic reporting

See general comments.

Experimental design

See general comments.

Validity of the findings

See general comments.

Additional comments

This is my second review of this paper. The authors have made a bona fide attempt to reply to the reviewers' comments, and I think the paper has notably improved as a result.

There is still an issue with a couple of the figure captions:

1) The caption of Fig. 2 still states in l. 119-120 that GMNH 4281 and NMR9991-00001783 are "right" mandibles and shown in "lateral" view. This is incorrect: both specimens are *left* mandibles and shown in *medial* view (this is obvious from the fact that the mandibular foramen, the symphysis, and the gingival foramina are all clearly visible).

2) The caption of Fig. 3 (l. 129) should specify that the periotic is shown in medial (left photo) and lateral (right photo) view. I still think that it would be better to assign individual labels (G and H) to these photos.

Otherwise, I have no further comments to make. Occasionally, modifications appear to have been carried out somewhat hastily, which means that the manuscript might benefit from another round of proofreading. For example, l. 94 states: "[...] f one dentary, three puted dentary elements [...].

In general, however, this paper is ready to be published.

---

## Round 0.3 · accepted · Accept

Congratulations. Please work with our production team to get your paper published.

#